# On the functional form of particle number size distributions: influence of particle source and meteorological variables

Katia Cugerone[1], Carlo De Michele[1], Antonio Ghezzi[1], Vorne Gianelle[2], and Stefania Gilardoni[3]

[1]Politecnico di Milano, Department of Civil and Environmental Engineering, Milano, Italy
[2]Regional Agency for Environmental Protection Lombardia, Milano, Italy
[3]National Research Council, Istitute of Atmospheric Science and Climate, Bologna, Italy

*Correspondence to:* Katia Cugerone (katia.cugerone@polimi.it)

**Abstract.** Particle number size distributions (PNSDs) have been collected periodically in the urban area of Milan, Italy, during 2011 and 2012 in winter and summer months. Moreover, comparable PNSD measurements were carried out in the rural mountain site of Oga-San Colombano (2250 m a.s.l.), Italy, during February 2005 and August 2011. The aerosol data have been measured through the use of Optical Particle Counters in the size range $0.3 - 25\ \mu$m, with a time resolution of one minute. The comparison of the PNSDs collected in the two sites has been done in terms of total number concentration, showing higher numbers in Milan (often exceeding $10^3$ cm$^{-3}$ in winter season) compared to Oga-San Colombano (not greater than $2 \cdot 10^2$ cm$^{-3}$), as expected. The skewness-kurtosis plane has been used in order to provide a synoptic view, and select the best distribution family describing the empirical PNSD pattern. The four-parameter Johnson SB (JSB) distribution has been tested for this aim, due to its great flexibility and ability of assuming different shapes. The PNSD pattern has been found to be generally invariant under site and season changes. Nevertheless, several PNSDs belonging to Milan winter season (generally more than 30 %) clearly deviate from the standard empirical pattern. The seasonal increase of the concentration of primary aerosols due to combustion processes in winter and the influence of weather variables, such as precipitation and wind speed, throughout the year, could be considered plausible explanations of PNSD dynamics.

## 1 Introduction

High suspended aerosol concentration in low atmosphere causes short-term health effects and increases the possibility of contracting serious chronic respiratory and cardiovascular diseases (Pope III and Dockery, 2006; Pope III et al., 2011; WHO, 2013). In addition, atmospheric aerosols reduce the visibility (Schwartz, 1996) and alter the Earth's radiation balance (Solomon et al., 2007; Stocker et al., 2013) are two additional environmental issues. In particular, aerosol size distribution is a climate relevant variable, able to modify the optical properties of particles and their cloud forming potential as well (Stocker et al., 2013)

Number concentrations of aerosol particles are generally very high in urban and kerbside environments, compared to rural and pristine areas, due to the proximity to pollution sources (Van Dingenen et al., 2004). Urban particle number concentration (in the range 10 nm - 800 nm) is usually above $10^4$ cm$^{-3}$ during the cold season, while at regional background site it do not exceed $10^3$ cm$^{-3}$ (Van Dingenen et al., 2004; Asmi et al., 2011). In urban environment the most important sources of

atmospheric aerosols are combustion processes (mainly related to traffic, residential heating, and energy production), road dust re-suspension, and formation of secondary aerosols from gas and particulate phase precursors. While combustion emissions and secondary aerosol contributes mainly to fine particles (below 1 $\mu$m), road re-suspension and part of the traffic emissions contribute to the coarse particle mode (Seinfeld and Pandis, 2006; Fuzzi et al., 2015). In pristine areas aerosols are mainly composed by secondary aerosol (fine particles) and re-suspended soil dust (coarse particles). Source and removal processes determine season and spatial variability of particle size distribution.

Models can describe particle number size distribution by two different approaches. In the sectional approach, particles are distributed into size bins, or sections. Such approach is computationally expensive, but does not require any assumptions about the functional form of the particle size distribution. More computationally efficient models represent aerosol with a limited number of modes, described by mathematical equations. This scheme requires to know the mathematical function that better describes the multimodal distribution of ambient aerosol. In the literature, different probability distribution functions have been used to represent particulate size distributions. The first one is the classic normal distribution, that was soon discarded because of its symmetry (Liu and Liu, 1994). Other distributions have been used in limited specific applications: Deirmendjian (1964, 1969) proposed the modified-gamma distribution for describing the particulate size distribution of marine or coastal particles for studying the light scattering phenomena; Brown and Wohletz (1995) employed the Weibull distribution to fit the aerosols generated from fragmented rocks; Rosin and Rammler (1933) introduced a new mathematical form, derived from the Weibull distribution, the so called Rosin-Rammler, to model the evolution of atmospheric aerosol. Later, Barndorff-Nielsen (1977, 1978) introduced firstly the hyperbolic, and secondly the generalized hyperbolic distribution to represent the statistical variability of sand grading. Other authors (Junge, 1963; Clark and Whitby, 1967; Pruppacher and Klett, 1980) proposed the power-law distribution to model atmospheric aerosol number size distribution. This is mathematically simple to compute, compared to other used functional forms, but it is accurate only over a limited size range, beyond which significant errors can arise (Leaitch and Isaac, 1991). Above all, the most used distribution to describe PNSDs is the lognormal (Jaenicke and Davies (1976); Whitby (1978), among others). This functional form is mathematically simple, easy to apply and allows a good match with a wide variety of empirical data, even if a theoretical justification for its widespread use does not exist, neither it has been identified to be superior to others in a general sense (Seinfeld and Pandis, 2006; Hinds, 2012).

Here, the four-parameter Johnson SB distribution is presented for modelling PNSD data, in addition to the classical mixture of (two) lognormals, widely used in the literature (Butcher and Charlson, 1972; Hinds, 1982; Seinfeld and Pandis, 2006). The great flexibility and versatility of this distribution, together with its boundedness (which matches the physical limitations of the analysed aerosol particles) make the Johnson SB a good candidate for this purpose. The use of this distribution has also been inspired by Cugerone and De Michele (2015); D'Adderio et al. (2016); Cugerone and De Michele (2017), where the Authors have recently demonstrated the accuracy of this probability function in modelling the number size distribution of drops (DSD) at the ground, a particular case of PNSD. Furthermore, the outcomes of this study are in accordance with the works of Yu and Standish (1990) and Liu and Liu (1994), in which JSB was firstly proposed for this aim. In order to statistically prove the adequacy of this distribution, we use the skewness-kurtosis moment ratio diagram (S-K plane), the locus of the couples skewness ($\beta_3$) and kurtosis ($\beta_4$), as diagnostic tool (Vargo et al., 2010). The availability of a great number of PNSD data makes

possible to plot a great amount of empirical couples $(\beta_3, \beta_4)$ on S-K plane. In this way, the general PNSD empirical pattern and in parallel a consistent theoretical distribution family can be individuated.

In the next, we present the results of the analysis of a great amount of PNSD data, collected with Optical Particle Counters in two different sites: the urban site of Milan and the mountain rural site of Oga-San Colombano. For these cases, the PNSD pattern (i.e. the domain where the sample points are in the skewness-kurtosis plane) is well represented by the Johnson SB domain, except for the urban winter data. In other words, the PNSD pattern does not change, even if we change the site or the season with the exception mentioned above. In winter, the pattern seems to be altered, probably by the influence of the aerosols punctual sources, which cause an increase of the total particle counts. In order to give our interpretation of these trends, we used the S-K plane to summarize, from the statistical point of view, the aerosol dynamics due to anthropogenic and also meteorological forcings.

## 2    Datasets and instrumentation

The data have been collected in two sites: the urban site located in Milan at Pascal-Città Studi (45°28'42", 9°13'54"E, 120 m a.s.l.) and the rural high altitude site at Oga-San Colombano (46°27'40"N, 10°18'07"E, 2290 m a.s.l.). The urban site is representative of "urban background" conditions and is not direct affected by local traffic emissions (Vecchi et al. (2004)), while the latter was a temporary experimental high altitude site. Tab. 1 provides details about the period of observation considered in this analysis and the number of minutes. We would like to point out that the analysed time ranges have been selected also according to the availability of the aerosol PNSD datasets, which were kindly provided by ARPA Lombardia.

Aerosol particle number concentrations were measured using an Optical Particle Counter (OPC), Grimm 107 Environcheck model, with a time resolution of 1 minute. The measured size distributions range from 0.3 to 25 $\mu$m subdivided in 26 size bins for all the analysed months and sites. The only exception is the winter dataset of Oga-San Colombano collected in February 2005 (SC1), characterized by size distributions between 0.3 and 25 $\mu$m, subdivided in 15 size bins. The OPC measurements allow the quantification of a portion of fine mode particles (between 300 nm and 1 $\mu$m) and the coarse mode particles (above 1 $\mu$m). The instrument is based on the quantification of the 90° scattering of light by aerosol particles.

The two sites present very different characteristics regarding the magnitude, the distribution and the composition of the aerosol fraction and the climatic characteristics. In particular, Oga San Colombano shows a higher relative contribution of organic aerosol, likely of secondary origin, as suggested by a higher organic to elemental carbon ratio Sandrini et al. (2014). Milano shows a higher nitrate to sulfate ratio, in agreement with a stronger impact from combustion sources, such as traffic and industrial emissions Perrone (2012). Milan has a humid subtropical climate (Cfa), according to the Köppen climate classification (Kottek et al., 2006). Milan's climate, as the all Valpadana valley, the Northern Italy's inland plain, is influenced by the natural barrier of the mountains (the Alps in the North and the Apennines in the South) which obstruct and prevent inflows from North, South and West. Winters and summers are usually dominated by high pressure, while autumns and springs are characterized by alternation between high and low pressure. These conditions cause usually high moisture levels in the low atmosphere and air stagnation, especially during high pressure seasons. Furthermore, Milan climatic conditions can be considered a typical

example of urban climate: urbanization has evidently changed the form of the landscape, and has also produced changes in the area's air. According to the European directive 2008/50/CE, all the European Countries must respect the standard limits related to the air quality (European Commission, 2008). In particular, the maximum daily levels of PM10 do not have to exceed 50 $\mu$g/m$^3$ (for more than 35 days per year), while the annual limit is 40 $\mu$g/m$^3$ (which should not be overcome on a yearly

average). During 2011 (2012), the maximum daily limit in Milan was passed 122 (97) times, during the winter period, and the average yearly level was 47 (43) $\mu$g/m$^3$.

On the contrary, San Colombano is a mountainous rural site, characterized by the typical Alpine climate (Dfb, according to the Köppen climate classification) with warm summers and long, cold and snowy winters. In the specific, the analysed site is located far away from higher mountains, and for this reason is not often shaded. Typically, San Colombano presents free air

circulation and stagnation of cold air during winters. These characteristics allow low aerosol levels (the average annual value of PM10 is 6 $\mu$g/m$^3$ with standard deviation of 5 $\mu$g/m$^3$), mostly consisting in particles produced far away and transported locally by the wind and thus good air quality all over the year.

The influence of primary aerosol sources and meteorology on PNSD has been investigated for the site of Milan. To study the effect of pollutant concentration here, we use nitrogen dioxide (NO$_2$) and nitrogen oxide (NO) measurements, collected

with a chemiluminescence technique following the requirements of European Standard EN 14211:2005:Ambient Air Quality. While, meteorological variables, namely precipitation and wind speed, have been measured respectively with a tipping bucket

**Table 1.** List of sites with indication of season, date of measurement and number of minutes.

| Code | Season | Date | N° minutes | Site |
|------|--------|------|------------|------|
| MI1 | Winter | January 2011 | 43409 | |
| MI2 | Winter | February 2011 | 21777 | |
| MI3 | Winter | February 2012 | 29221 | |
| MI4 | Winter | January 2014 | 41760 | Milan |
| MI5 | Summer | July 2011 | 36621 | |
| MI6 | Summer | August 2011 | 28715 | |
| MI7 | Summer | July 2012 | 42833 | |
| MI8 | Summer | August 2012 | 41945 | |
| SC1 | Winter | February 2005 | 40320 | Oga |
| SC2 | Summer | August 2011 | 44243 | |

rain-gauge and by an anemometer located at the ARPA station of Lambrate (45°29'46''N, 9°15'28''E, 120 m a.s.l.), which is around 3 km away from Pascal-Città Studi.

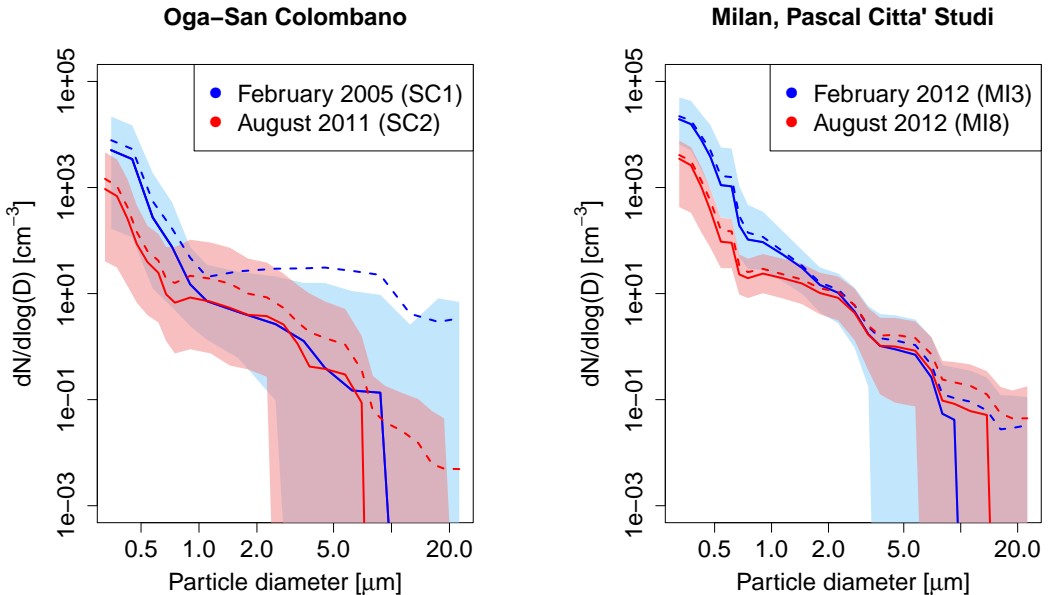

**Figure 1.** Median (solid lines) and mean (dashed lines) number size distributions for two selected winter and summer datasets of Pascal-Città Studi (MI3, MI8) and the two datasets of Oga-San Colombano (SC1, SC2). The coloured blue and red areas are limited below and above by the 5[th] and the 95[th] percentiles of the correspondent dataset.

## 3   Data analyses

The different characteristics of the aerosol number size distributions at Milan (urban site) and Oga-San Colombano (rural site) are shown in Fig. 1. Here, median (solid lines) and mean (dashed lines) number size distributions for the two datasets of Oga-San Colombano (SC1, SC2 in the left panel) and two selected winter and summer datasets of Milan (MI3, MI8 in the right panel) are shown, together with the 95-confidence interval (coloured areas). Similar results can be obtained considering the other datasets of Milan. Median and mean number size distributions of all the datasets present a common decreasing trend inside the analysed size range, 0.3-25 $\mu$m, but also evident discrepancies. In particular, the coloured areas depicted in the plots denote a greater PNSD variability for SC1 and SC2 compared to MI3 and MI8. The distance between the 5[th] and the 95[th] percentiles of the latter is very small in the diameter range 0.3-3 $\mu$m, accumulation mode particles and smaller coarse mode particles, and is characterized by higher number concentrations (especially in winter) compared to Oga-San Colombano.

Looking at these plots, it is difficult to understand if a unique functional form is able to describe the variety of PNSDs. In order to clarify this issue, we propose the use of the skewness-kurtosis moment-ratio diagram as diagnostic tool. This plane

was introduced by Craig (1936) and then updated by various authors, e.g. Balakrishnan et al. (1994); Cugerone and De Michele (2017). The skewness-kurtosis ($\beta_3 - \beta_4$) plane presents the skewness (Eq. 1), in abscissa, and the kurtosis (Eq. 2), in ordinate,

$$\beta_3 = E\left[\left(\frac{X - \mu_X}{\sigma_X}\right)^3\right] \tag{1}$$

$\quad \beta_4 = E\left[\left(\frac{X - \mu_X}{\sigma_X}\right)^4\right] \tag{2}$

where $X$ is the variable, $E[.]$ is the expected value, $\mu_X$ and $\sigma_X$ are respectively the mean and the standard deviation of $X$. The Pearson limit curve $\beta_4 - \beta_3^2 - 1 \geq 0$ (Pearson, 1916) divides the theoretically impossible and possible areas, in which a couple ($\beta_3, \beta_4$) can be found. In this diagram, each theoretical probability distribution is represented by a domain, which can be a point, or a line, or an area, depending on the number of shape parameters involved. Therefore, this plane can be used as

$\quad$ a diagnostic tool for the identification of distributions able to model given datasets, comparing the theoretical domain of the distributions and the sample variability of data. In Fig. 2, following Cugerone and De Michele (2015, 2017), we have reported the domain of some families of distributions including: normal, exponential, gamma, lognormal, Johnson SB and Johnson SU, and in addition, for the first time, a mixture of two lognormals. From Figure 2, it is possible to identify the following features:

– normal and exponential are represented by a single point. In particular, the square (0,3) represents the normal, and the
15 $\quad$ triangle (2,9) the exponential;

– gamma (long-dashed line) and lognormal (dotted-dashed line) are distributions represented by a line;

– the Johnson SB is the upper and lower bounded family and occupies the area (medium grey region) limited below by the Pearson limit curve and above by the lognormal line. The Johnson SU is the unlimited family; it covers all the rest of the plane (light grey region) and is limited below by the log-normal curve

$\quad$ – the domain of a mixture of two lognormals (red dots area) is represented by an area embracing the lognormal line. The domain has been determined numerically by Montecarlo simulations as reported in Appendix A.

Given this, we use the skewness-kurtosis plane to search the best functional forms able to describe the sample PNSDs of Milan and Oga-San Colombano datasets and to analyse the effects of primary pollutants and weather variables, in order to clarify the dynamics of PNSD variations.

$\quad$ **4 Results and discussions**

**4.1 Urban vs rural sites**

In order to analyse and compare PNSDs characterized by different ambient conditions and different seasons, we calculate the skewness and the kurtosis of each minute of the four datasets previously considered, SC1 (a), SC2 (b), MI3 (c), MI8 (d), and

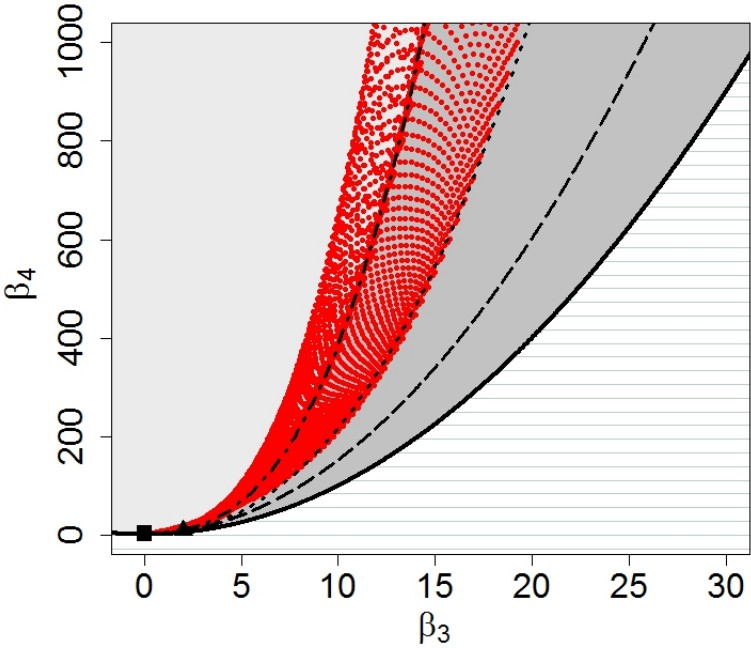

**Figure 2.** $(\beta_3, \beta_4)$ domain of some families of distributions including: normal (square), exponential (triangle), gamma (dashed line), lognormal (dotted-dashed line), Johnson SB (medium grey area) and Johnson SU (light grey area), and a mixture of two lognormals (red dotted area).

we report the sample couples $(\beta_3, \beta_4)$ in the skewness-kurtosis plane, see Fig. 3. In this plane, the limit curve represented by the tick black solid line, divides the statistically unfeasible area (dashed area) by the portion of the plain that can be occupied by the pairs $(\beta_3, \beta_4)$. The theoretical domains of some selected distributions are also reported: normal (black square), exponential (black triangle), lognormal (black dotted-dashed line), Weibull (black dotted line), gamma (black dashed line), Johnson SB,

(JSB - dark grey area) and Johnson SU (light grey area). The domain of the mixture of two lognormal (not reported here), being characterized by a greater shape variability respect to the simple lognormal distribution, is represented by an area embracing the lognormal line (see Appendix A for more detail). The sample couples $(\beta_3, \beta_4)$ are divided in four classes and coloured as function of the total particle count (TP) of the related minute: cyan dots represent PNSDs with TP<25000, blue dots 25000<TP<62500, purple dots 62500<TP<100000 and magenta dots TP>100000.

Firstly, by analysing Fig. 3, we see that the great part of the couples are located inside JSB theoretical domain, see Tab 2. This is a four-parameter distribution characterized by a bounded domain, which is suitable for the particles diameter, being a finite variable. MI3, the Pascal-Città Studi dataset collected on February in midwinter, represents the only evident exception with the 29.6% of the data points outside the JSB domain. Fig. 3 indicates also the existence of a relation between the position of the dots and their colour, and thus between the functional form and the total particle count. In particular, the sample skewness-

kurtosis couples tend to move left and to exit from the JSB domain with the increase of TP. This is again specially true for MI3,

which is characterized by the 96% percentage of the data points belonging to the fourth class, with TP>100000, while for SC1, SC2 and MI3 the percentages are respectively 18%, 2% and 14%. To stress this point, let's consider only the data outside the JSB domain. The percentage of data outside JSB with TP>100000 is 73.2%, 33.3%, 99.5% and 95.5%, respectively for SC1, SC2, MI3 and MI8. The smaller percentage observed for SC2 is likely due to the limited number of data points characterized by large TP, and thus not statistically significant. We can conclude that, in case of very high number of aerosol particles, the Johnson SB cannot be considered anymore the most accurate distribution in describing PNSDs. From Fig. 3, it is possible to see that, the data points outside the JSB domain, are located within the domain of Johnson SU (light grey) and in some cases in the domain of the mixture of two lognormals, which seem good candidates to represent them, even if these distributions are

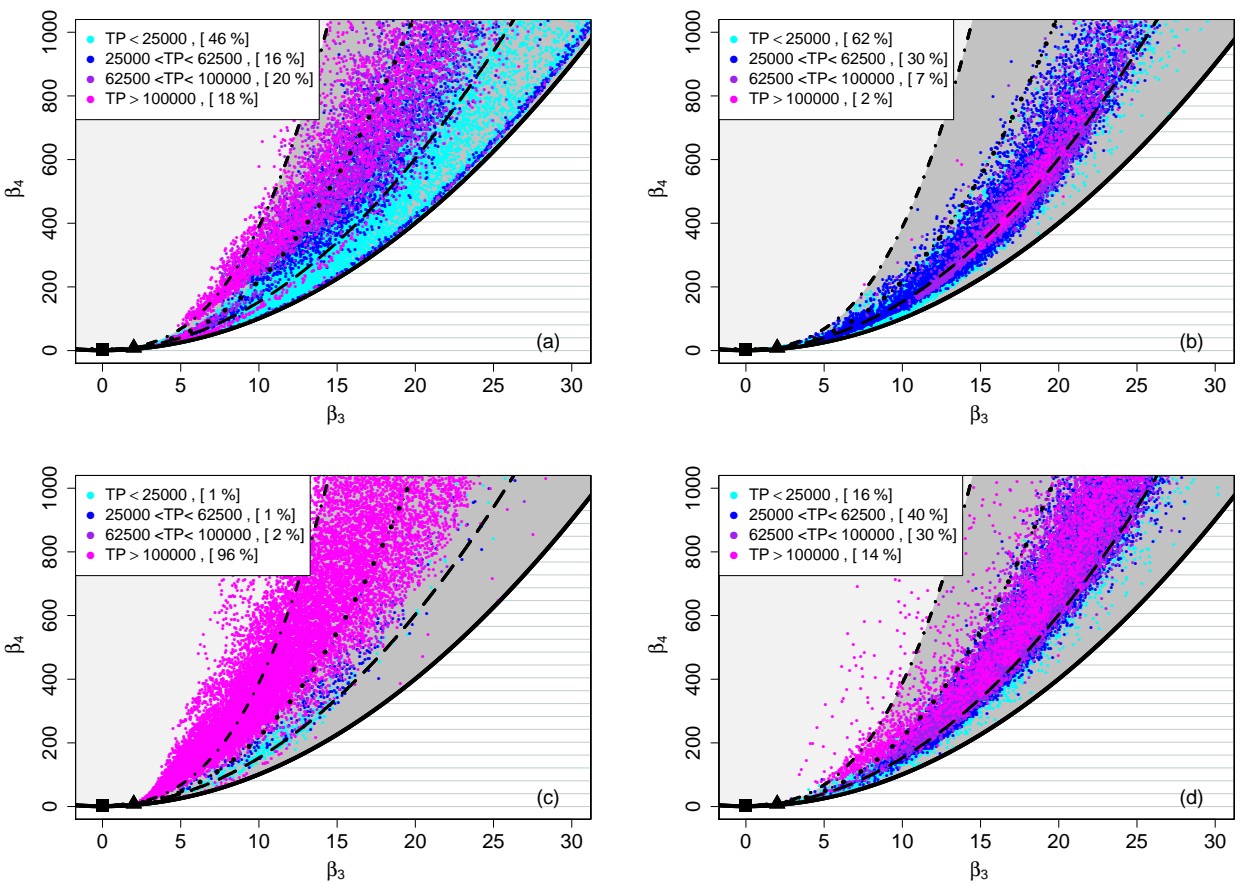

**Figure 3.** Location of the sample couples ($\beta_3$,$\beta_4$) from SC1 (a), SC2 (b), MI3 (c), MI8 (d) datasets in the skewness-kurtosis plane with theoretical distribution reference domains. Cyan dots represent PNSDs with TP<25000, blue dots 25000<TP<62500, purple dots 62500<TP<100000 and magenta dots TP>100000. The dotted-dashed line is the lognormal distribution, the dotted line is the Weibull distribution, and the dashed line is the gamma distribution.

not upper bounded, like the variable under investigation. Similar results have been obtained for the others Milan datasets, see Appendix B.

**Table 2.** Quantitative analysis of Fig. 3

| $(\beta_3,\beta_4)$ | SC1 [%] | SC2 [%] | MI3 [%] | MI8 [%] |
|---|---|---|---|---|
| with TP$>10^5$ | 18 | 2 | 96 | 14 |
| outside JSB | 1.1 | 0 | 29.6 | 0.4 |
| with TP$>10^5$ outside JSB | 73.2 | 33.3 | 99.5 | 95.5 |

As proof of this, we have fitted the JSB distribution to the measured PNSDs, using the Maximum Likelihood Method (see Appendix C for more details). Fig. 4 shows the size distribution of particle number at the urban site MI3 in winter (red), at the urban site in summer MI8 (blue), and at the rural site SC2 (green), together with their corresponding position in the $(\beta_3,\beta_4)$ space. The three PNSDs are representative of their own datasets. In particular, the MI3 PNSD collected at 9:57 pm (UTC) counts 1155778 particles, the MI8 PNSD collected at 4:34 pm (UTC) counts 9465 particles and the SC2 PNSD collected at 8:41 pm (UTC) counts 2755 particles. The JSB accurately fits PNSDs data in the last two cases, characterized by a lower total number of particles. The figure clearly illustrates that when the size distribution is efficiently described by the JSB parameterization, the corresponding distribution falls in the grey area, the JSB domain. The mixed lognormal distribution, represented in the skewness-kurtosis plane with dark grey dots, is not able to represent the PNSDs in none of the three cases.

## 4.2 Primary pollutants influence

The analysis of the skewness-kurtosis plane shows the existence of a PNSD pattern, which is generally under site and season changes, with the exception of the winter datasets. During winter seasons in urban environments, a significant change of the general PNSD pattern, consisting in a shift toward the centre of S-K plane, has indeed been observed. A plausible explanation of PNSD dynamics can be found in the recurrent winter increase of aerosol emissions (much more evident in urban sites), due to heating ignition and high traffic levels.

In order to clarify this point, measurements of two common atmospheric components, in particular nitrogen dioxide ($NO_2$) and nitrogen oxide (NO) collected at Pascal-Città Studi, have been considered. Nitrogen oxides ($NO_x$=NO+$NO_2$) can form naturally in the atmosphere by lightning and some is produced by plants, soil and water, but their major source in urban areas is the burning of fossil fuels, like coal, oil and gas (World Health Organization, 2000). $NO_x$ are mainly produced by combustion processes. Nevertheless primary combustion emissions are dominated by NO over $NO_2$. $NO_2$ can then be formed through the oxidation of NO in the atmosphere. It follows that the $NO_2$ to $NO_x$ ratio can provide a measure of the oxidative capacity of the atmosphere (Rao and George, 2014; Fernández-Guisuraga et al., 2016) and it is a measures of the temporal proximity to

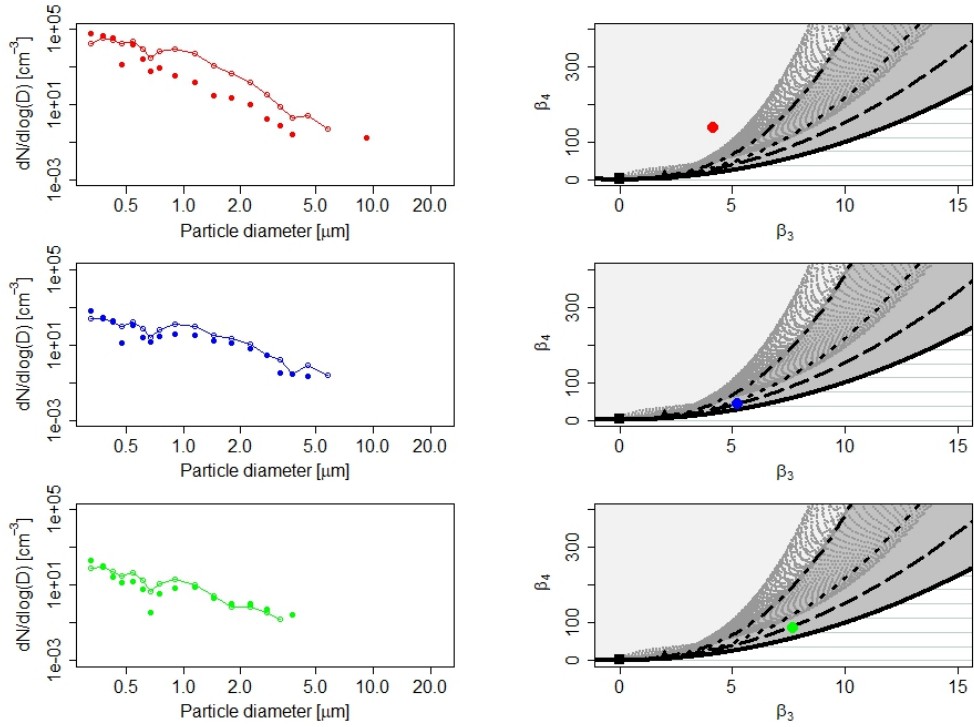

**Figure 4.** Comparison of empirical PNSDs (dots) and JSB fittings (lines) from MI3 (red), MI8 (blue) and SC2 (green), first column. Correspondent location of the $(\beta_3, \beta_4)$ couples in the (S-K) plane, second column.

emission sources. In addition, measurements performed in Milan during different field experiments show that the ratio of $NO_2$ to $NO_x$ anti-correlates with the ratio of black carbon to PM1, suggesting that the $NO_2/NO_x$ in these urban areas is an indicator for secondary pollutant formation relative to primary traffic emissions. In Fig. 5 we have again reported the skewness-kurtosis plane, where we have plotted in black the data points of MI1 (upper panel) and MI2 (lower panel). Then, we have selected the data points belonging to minutes characterized by values of the ratio $NO_x/NO_2$ between 1 and 1.1 (red dots - strong prevalence of secondary aerosols), 1.1 and 1.5 (orange dots - ligh prevalence of secondary aerosols), 1.5 and 3 (yellow - light prevalence of primary aerosols), greater than 3 (green - strong prevalence of primary aerosols). Both the two datasets are characterized by high aerosol numbers and high percentages of data points outside JSB domain (74 % and 65% respectively). The percentages of data points characterized by a ratio $NO_x/NO_2$ greater than 3 are around 50 %, indicating a prevalence of the primary aerosol contribution. Most of such data points fall in the region characterized by a size distribution dominated by sub-micron particles as depicted in Fig. 4. If we select only the data points outside the JSB domain, the percentage of data points with ratio greater than 3 (strong prevalence of primary aerosols) is 56 % for MI1 and 50 % for MI2. While, the percentage of data points with ratio greater than 1.5 (light or strong prevalence of primary aerosols) is 88 % for MI1 and 67 % for MI2. These findings support our hypothesis that in urban sites during winter season the increase of primary aerosols emission by local sources causes an

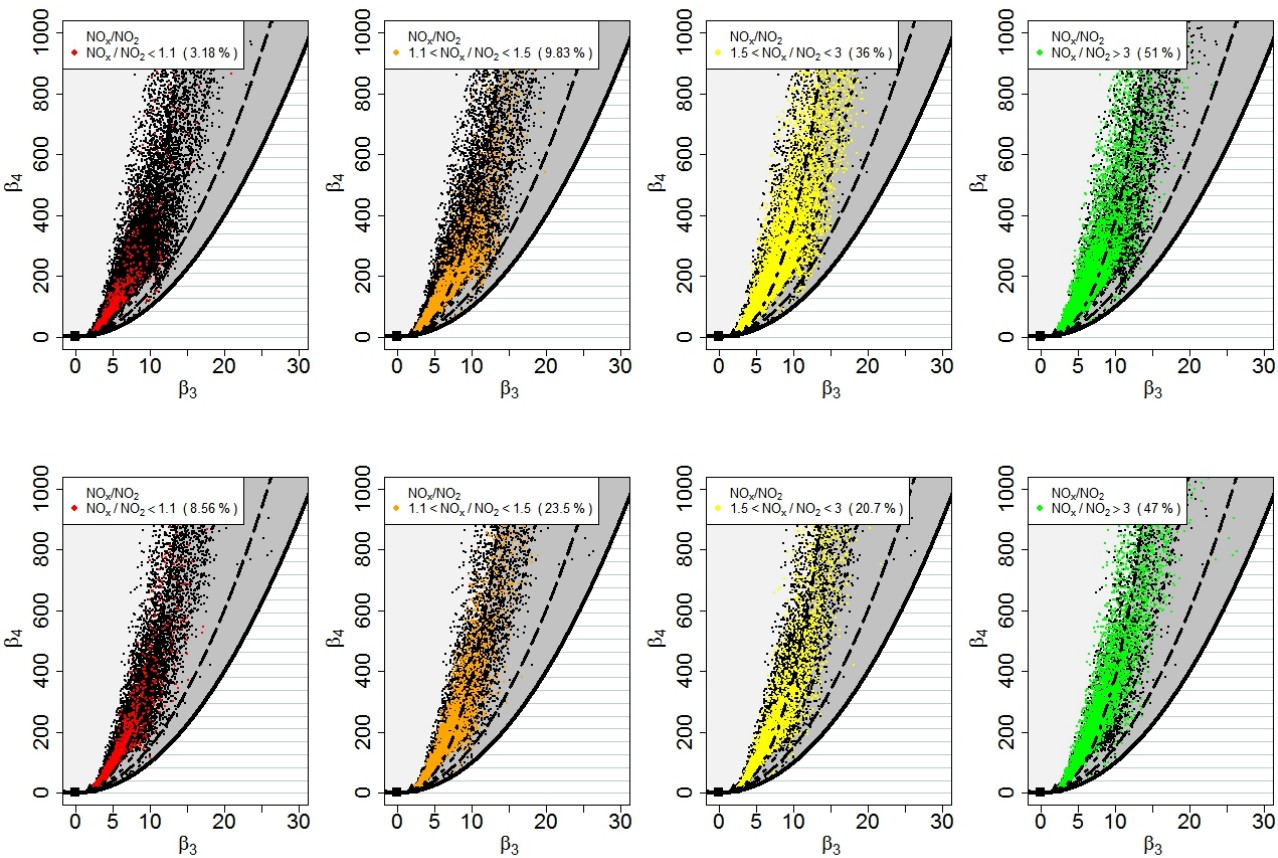

**Figure 5.** Location of the sample couples ($\beta_3$,$\beta_4$), black dots, from MI1 (upper panel) and MI2 (lower panel) in the skewness-kurtosis plane with theoretical distribution reference domains. Red dots represent PNSDs with ratio NOx/NO$_2$ between 1 and 1.1, orange dots 1.1<NOx/NO$_2$<1.5, yellow dots 1.5<NOx/NO$_2$<3 and green dots NOx/NO$_2$>3.

evident increase of primary aerosol compounds concentration. This can be considered one of the cause of the location shifts of ($\beta_3$,$\beta_4$) couples in the skewness-kurtosis plane.

### 4.3 Weather variables influence

The aerosol concentration variation due to the increase of particles emissions in the atmosphere is not the only cause of PNSD shape changes. The occurrence of weather events, such as precipitation or high wind speed, indirectly causes a modification of the particle concentration, which often results in a decrease of number and mass of aerosol particles suspended in the low atmosphere (Cugerone et al., 2018). In particular, the aerosol wet removal during rain events, known as scavenging process, is caused by the vertical movement of the falling raindrops, which intercept the suspended aerosol particles and bring them to the ground (Seinfeld and Pandis, 2006). While, the blowing of high wind speeds in urban areas far away from the sea, like

Milan, cleans the atmosphere by dispersing and diluting the aerosol particles and preventing local accumulations (Harrison et al., 2001). The visible effect of these phenomena in the skewness-kurtosis plane is described again by a shift of the position of the couples ($\beta_3$,$\beta_4$), but in the opposite direction (toward the right-side of S-K plane) respect to the one (toward the centre of S-K plane) caused by the influence of the high concentration of primary aerosol compounds. In other words, the couples

($\beta_3$,$\beta_4$) are forced to stay in the JSB theoretical domain. Therefore, looking at the skewness-kurtosis plane, the influence of the weather variables is visible only if the occurrence of high wind speed or significant precipitation events are foregone by particle size concentrations characterized by couples ($\beta_3$,$\beta_4$) outside JSB domain. These conditions generally occur in winter seasons in urban areas, as we have seen in the previous paragraph.

Two practical examples are taken from the dataset MI3 (February 2012). The first is related to precipitation (Fig. 6), the

10 second to wind (Fig. 8). In both the two cases we represent the skewness-kurtosis plane and we track the movement of the couples ($\beta_3$,$\beta_4$) before, during and after the respective weather event. The dots are coloured according to the time window in which the PNSDs have been collected. Fig. 6 shows the location of the sample couples ($\beta_3$,$\beta_4$) in the skewness-kurtosis plane during the days 19-20-21 February 2012; each day is represented in a specific panel to visualize the movements of the couples precisely. On February 19[th] the couples are stably located outside the JSB domain, as normally for urban site in midwinter, no

precipitation occurs. In February 20[th] between 9am and 5pm, a rainfall event with a maximum intensity of 1 mm/h and average intensity of 0.6 mm/h occurred. The influence of meteorological change is clearly reflected in the skewness-kurtosis plane: the

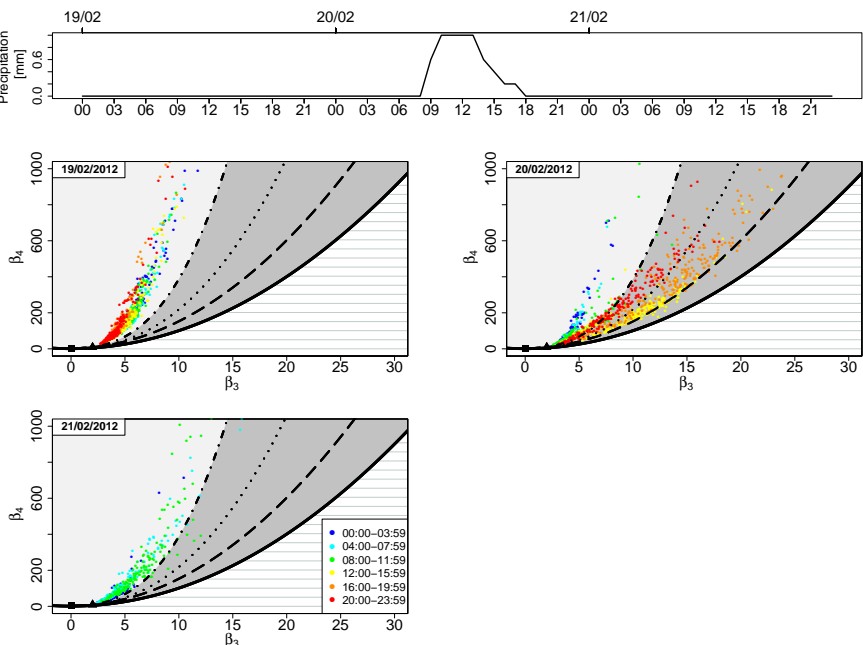

**Figure 6.** Location of the sample couples ($\beta_3$,$\beta_4$) in the skewness-kurtosis plane for the days 19-20-21/02/2012. The dots are coloured according to the time slot as explained in the legend. Between 9am and 5pm of February 20[th], a precipitation event with a maximum intensity of 1 mm/h and average intensity of 0.6 mm/h occurred.

dots related to PNSDs collected between midnight and 8 am (blue and cyan) on this day are still outside JSB area, but starting from 8 am until around 8 pm the dots (green, yellow and orange) move right and enter inside JSB domain, then after 8 pm

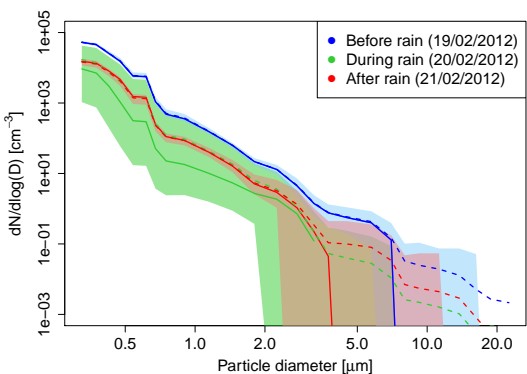

**Figure 7.** Median (solid lines) and mean (dashed lines) number size distributions for the minutes before (blue), during(green) and after (red) the rain event on February 20th 2012. The coloured blue, green and red areas are limited below and above by the 5[th] and the 95[th] percentiles of the correspondent dataset.

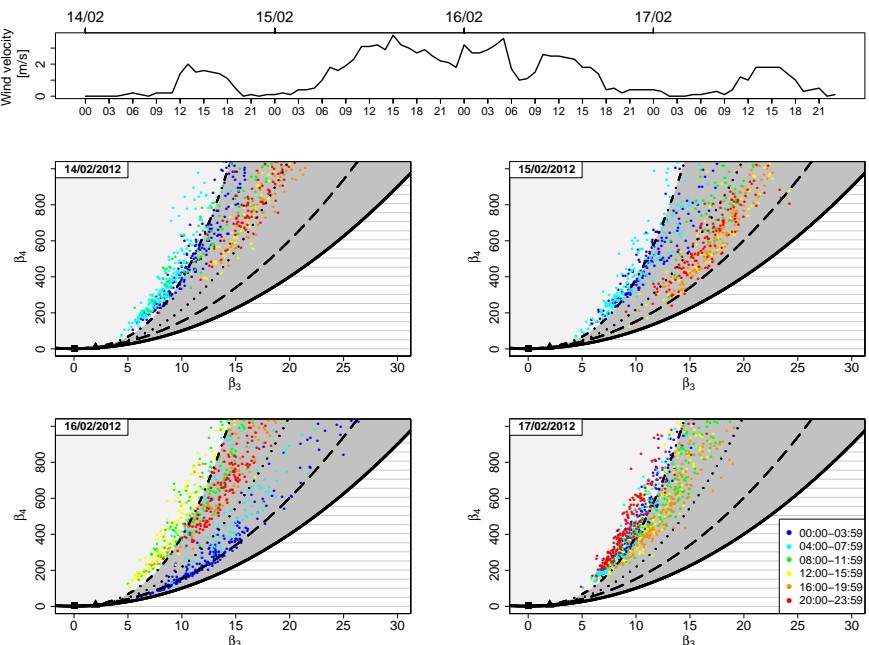

**Figure 8.** Location of the sample couples ($\beta_3$,$\beta_4$) in the skewness-kurtosis plane for the days 14-15-16-17/02/2012. The dots are coloured according to the time slot as explained in the legend. Between February 15[th] and February 16[th], relatively high wind speed with maximum of 3.8 m/s (recorded on 15/02 at 3pm) occurred.

they start to exit (red dots) and remain outside also during February 21$^{th}$. In Fig 7, median (solid lines) and mean (dashed lines) number size distributions measured in the minutes before (blue), during (green) and after (red) the rain event are shown, together with the 95-confidence interval (coloured areas). The distribution during and after the rain event are characterized by lower aerosol concentration levels respect to the distribution before the rain. This phenomenon is a clear consequence of the scavenging effect of precipitation. Furthermore, in the distribution before the rain event, the high particle concentration in the smaller diameter classes and the existence of outliers with diameters bigger than 4-5 $\mu$m can be considered the reason of the position outside the JSB domain in the skewness-kurtosis plane. Distributions having such strong peak in the first diameter classes, but with the presence of particles in big diameter classes, can be generally represented by theoretical distributions having a 'long' right tail, characteristic of unbounded distributions.

Similarly, the effect of high wind speed is reported in Fig. 8, where the location of the sample couples ($\beta_3$,$\beta_4$) in the skewness-kurtosis plane during the days 14-15-16-17 February 2012 is shown. During these days, and in particular between February 15$^{th}$ and February 16$^{th}$, relatively high wind speed with maximum of 3.8 m/s (recorded on 15/02 at 3pm) occurred. The increase of wind speed causes a movement to the right (inside JSB area) of the couples ($\beta_3$,$\beta_4$), but when the wind speed decreases the couples returns outside JSB domain fairly quickly.

In other words, the modifications of PNSDs shape (caused by increase or decrease of aerosol concentration by new emissions or weather events) result in movements of the sample ($\beta_3$,$\beta_4$) couples in the moment ratio diagram, and consequently in changes of the pool of distributions potentially able to describe the PNSDs. Inside this pool, the Johnson SB distribution seems the best, in case of low polluted conditions - achievable also after significant weather events, such as precipitation or high wind speed. Gamma, lognormal and Weibull can be considered accurate too, but for a limited number of times. Conversely, in case of high concentration, none of these distributions should be inserted into the pool: the PNSD shape changes because of the elevated increment of the fine particles, which shift the sample mode to the left and cause an increase of the number of outliers. ($\beta_3$,$\beta_4$) couples are shifted into an area of the skewness-kurtosis plane belonging to unlimited distributions, such as the Johnson SU or the Generalized Hyperbolic. These distributions become the best suitable for PNSDs characterized by this kind of shape, despite the limited (above and below) nature of the variable under exam, the aerosol diameter.

## 5   Conclusions

In this work, OPC data, collected in two sites, one urban, Milan at Pascal-Città Studi, and one rural, Oga-San Colombano, have been analysed. The aerosol particle number concentrations present very different characteristics: the magnitude, the composition of the aerosol fractions and also the empirical distribution shape vary a lot within the dataset. These variations are caused both by the different nature of the measurement site (urban and polluted Pascal-Città Studi, rural Oga-San Colombano) and by the season of measurement. This likely suggests that a unique statistical distribution cannot be able to cover all of the PNSD forms.

In order to statistically identify the best distribution describing the empirical PNSD pattern, the skewness-kurtosis plane has been used as a synoptic tool. Our analyses show that the four-parameter Johnson SB, thanks to its flexibility, could be

considered the most accurate distribution for representing the empirical PNSD forms, being able to describe the great majority of the PNSDs analysed, under conditions of low pollution level. Other distributions, such as gamma, Weibull, lognormal and mixture of two lognormals, could be considered adequate too, but for a much more limited portion of the datasets. Further work is needed to quantitatively describe the goodness of the JSB distribution, and the link between the fit parameters and aerosol sources and processes. Such analyses will be the subject of a future study.

The urban datasets, collected on January and February in midwinter, represent the only evident exceptions, since their PNSDs clearly deviate from the pattern generally found for the other datasets. The combined analysis of the skewness-kurtosis plane and the ratio between NOx and $NO_2$ suggested that the increase of the concentration of primary aerosol compounds, due to the seasonal winter increase of combustion processes from a countless number of punctual sources, can be considered a plausible explanation of PNSD empirical pattern dynamics. In particular, the increment of the fine particles shifts the sample mode to the left and causes an increase of the number of outliers. In these cases, distributions like the Johnson SU and the mixture of two lognormals can be good candidates. This important issue will be investigated in future works.

Nevertheless, also the occurrence of weather events, such as precipitation and high wind speed, by indirectly decreasing the aerosol concentrations, can have considerable influence in PNSD dynamics. The influence of these events is particularly evident in winter months, when the aerosol concentration is normally high and the PNSDs deviate from the general empirical pattern. Precipitation or high wind speed decrease the aerosol concentration, making the empirical PNSDs following again the general empirical pattern, even if for a limited time (more or less corresponding to the duration of the event itself).

*Acknowledgements.* The authors want to thanks ARPA Lombardia, Settore Monitoraggi Ambientali, for the access to OPC and gas-phase composition data.

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
