# Peer review of "On the functional form of particle number size distributions: influence of particle source and meteorological variables"

_Atmospheric Chemistry and Physics, 2016_

## Referee Comment (RC1) · Anonymous Referee #3 · 18 May 2017

General – The authors examine particle size distributions from about 0.3 um to 5 um diameter using two optical particle counters (OPC) located in Milan and on Oga-San Colombano at 2250 msl. They use a skewness-kurtosis plane based on the statistics of the measured size distributions to consider several possible statistical distributions that are commonly used to represent aerosol particle size distributions. They show that the Johnson SB (and SU) are capable of representing relatively complex distributions over the size range of the OPC (0.3-5 um) depending on factors that control total number concentrations, including wind speed and precipitation.

This paper offers a new (to me) and interesting approach to contrasting particle size distributions, although it would have been more interesting had the size distributions

covered sizes down to 20 or 30 nm, rather than starting at 300 nm. The work appears to be targeted towards models that use modal representations, but the skewness-kurtosis plane with separation by total number concentration seems like it has potential as a tool for analysis of size distributions.

The paper is well organized, and the figures are well done, but a careful editing of the paper for grammar is required as misinterpretations are possible. The figure captions could use some additional detail.

Specific comments:

1. Page 3, lines 4-5 – You first say the PNSD pattern does not change, and then you say that it varies by site and season. Please clarify. Also, clarify "In this case": winter?

2. Page 3, line 10 – What do you mean by "background"? Its meaning in this context needs to be clarified.

3. Page 4, line 1 – What model number? Did you use two counters, one at each site, or was one counter transported between sites? If two counters, were they the same model at both sites? If two counters, how were they compared and validated to determine possible differences associated with the counters rather than the sites? It is not uncommon for these types of counters to have large uncertainty in the smallest nominal size. Was the lowest size evaluated in any way?

4. Page 4, lines 17 and 19 – do you mean ug/m3?

5. Page 4, line 20 – Similar to 'background', how do you define pristine?

6. Page 4, Line 23 – What are low aerosol levels?

7. Page 4, lines 25-29 – This tells us nothing other than you have measured some other things (NOx and meteorological quantities). Perhaps that is what you intended, and there is simply a grammar issue? Otherwise, is there a reference for the "influence" investigation? NOx may be considered a component of the aerosol, but it is not an

"aerosol compound".

8. Figure 1 – Why did you choose August 2012 (MI8) rather than August 2011 (MI6) that is at the same time as SC2?

9. Page 5, lines 6-8 - Figure 1 – The inflections in the distributions between about 0.4 um and 0.8 um are present at both sites, but particularly evident in the Milan results. Rather than "droplet mode particles", the inflections may be a symptom of ambiguities in the scattering function for the particular angle of the OPC, sometimes referred to as Mie ambiguities. Some of the peaks and valleys above 1 um may also be due to this potential problem. The exaggerated inflections in the Milan PNSD relative to the SC PNSD may be due to differences in the index of refraction or the apparently less steep Milan distributions from 0.3 to 0.7 um compared with the SC distributions. You should look at the scattering function versus particle diameter for the counters. If ambiguities are present (i.e. a similar amount of light scattered into the collection angle by particles of different sizes), the common solution is to average across the bins covering the ambiguity size range.

10. Figure 2 – Indicate what the lines refer to in the caption.

11. Page 7, line 7 and Figure 3 – The fits in Figure 3 are very good considering the detail. Not being familiar with the JSB distribution, I would to see your fitting process discussed in a little more detail, including the Maximum Likelihood Method. The details could be added to the supplement.

12. Page 10, lines 10-17 – Higher values of NOx/NO2 may indicate closer temporal proximity to sources. They also suggest the possibility of a greater fraction of particles from primary emissions, but it does not guarantee that primary particle emissions dominate over secondary. Also, it sounds like "These findings support our hypothesis that in urban sites during winter season the increase of primary aerosols emission by local sources causes an evident increase of primary aerosol compounds concentration." is saying that an increase of primary aerosol emissions causes an increase in primary

aerosol concentrations. I presume that is not exactly what you intended. I suggest saying something like particle concentrations increase when sampling is done in closer photochemical proximity to sources. Remove references to primary or secondary; they are not needed.

---

## Referee Comment (RC2) · Anonymous Referee #4 · 2 Jun 2017

This manuscript describes the applicability of the Johnson SB distribution for fitting particle size distributions measured by optical particle counters operated at two sites in Italy. The paper first focuses on assessment of the merits of the expression form by examining the fraction of measurements that lie within the region on the skewness-kurtosis plane that is bounded by the Johnson SB envelope of possible solutions. The patterns of data point clusters on that plane are then linked with meteorological and environmental conditions to suggest the more general use in describing the sources and processing responsible for an observed size distribution. The manuscript is reasonably well-written but would require some editing prior to publication.

[Figure]

I have identified several specific concerns I have with the manuscript below. More generally though, this simply does not seem to be appropriate for ACP. The dataset described is very limited and rather uninteresting when not complemented by other aerosol and trace gas measurements. More importantly, the dataset is not really the focus of the paper, but rather the technique to describe the dataset is. Thus, in its current form this would be more appropriate for a journal such as AMT. If the authors chose to shift the emphasis more toward the size distributions I still feel that because of the limitations of the dataset this would be better suited for another journal. It could be that collaboration with researchers involved in more comprehensive measurement campaigns could be valuable for evaluating the utility of the techniques described here for understanding influences on size distributions.

It seems the authors have considerable experience with statistical methods and data analysis, but not with air quality. The relevance of this is that there is far too much text describing rather fundamental details about aerosol sources and sinks and meteorology.

The averaged and example size distributions shown in Figures 1 and 3 reveal a common characteristic of distributions measured by OPCs - erroneous peaks and troughs that are often linked with features in the scattering intensity vs. size relationship for the optical geometry of the instrument. The fact that they are retained in the distributions suggests the authors didn't invest much time in calibration of the instruments and processing of the data. But more relevant for this paper, those features will influence any fit of the distributions and the location on the S-K diagrams. There is no discussion of these features or their impact.

The authors argue that the Johnson SB distribution is more appropriate for fitting the particle size distributions than more commonly used forms such as the lognormal. But they neglect to discuss the utility of the lognormal because of the direct connection of the parameters describing it with physically meaningful elements of the aerosol distribution (i.e., N, Dp_mean, SD) and the ability to describe variation of those parameters

accompanying things such as atmospheric processing. Furthermore, the manuscript largely dismisses lognormals based on the difference between the data points and the single lognormal point on the S-K diagrams. But does the representation as a point presume that only one lognormal is used to fit the distribution? In practice, multiple lognormals are almost always used.

Minor issues:

Page 4, line 1: Grimm model what?

Page 4, line 7: What is the basis for the assertion that the composition is different between the two sites. It undoubtedly is, but this still needs some support.

Page 4, line 26: Nitrogen dioxide and nitric oxide are not aerosol compounds.

Page 6, line 24: Total particle count is meaningless to readers. I assume the authors simply need to divide this by the product of flow rate and sample time to report it in concentration. Additionally, it seems there is confusion about the upper threshold value because it is written both as $10^4$ and as 100000 (=$10^5$).

Page 10, top: The NO2 to NOx ratio will be largely dependent on time of day, which will confound the interpretation of its influence on the patterns in the S-K diagrams.

Figure 3: The use of an unnecessarily large y-axis range obscures the information in the distributions and the quality of the fits.

Figure 4: The differences among these graphs are pretty modest.
* * *

---

## Referee Comment (RC3) · Anonymous Referee #5 · 3 Jun 2017

The manuscript proposes that the skewness-kurtosis plane (the size distribution projected into the third and fourth moments) can be used to follow changes to the particle number size distribution (PNSD), and that four-parameter Johnson SB (JSB) distribution as being sufficient for observing changes to the PNSD in a way that maps to the skewness-kurtosis plane. The authors present PNSDs from four measurement campaigns under different NOx and meteorology conditions. This manuscript includes a few interesting ideas that are less well-known to the broader ACP community, and has potential for novelty and impact. However, at present time the manuscript is strongly recommended for revision and re-submission. The reason for this is recommendation that each of these ideas introduced are not fully developed. As a result, the reader

is mostly left with an impression that what is demonstrated is that the size distribution changes when there are changes in meteorology or emission sources, which could be characterized more informatively using traditional approaches (number concentration, modes, etc.).

The conclusion that the JSB can be used to represent PNSDs does not appear to be well-supported by the material that is presented. As one of the other reviewer notes, PNSDs can be multimodal, and representing each of these modes well is in itself a challenge. There is no indication regarding the modality or quality of fit permitted by JSB. What is presented seems to be that the range of skewness and kurtosis in observed PNSDs fall within the range that can be represented by the JSB distribution except at high concentrations. Furthermore, it is not demonstrated that JSB outperforms other parametric distributions for representing PNSDs (except for the reason of having four fitting parameters), and the authors even note in the conclusions that the other parametric representations may be adequate.

Regarding the use of the skewness-kurtosis (S-K) plane, does it provide more information that cannot be achieved by examining other parameters of the PNSD conventionally used (e.g., first and second moments of lognormally transformed data)? Given the long history of modeling PNSDs, the mode gives some indication of whether the dominant source is likely anthropogenic or biogenic; the geometric standard deviation may be related to the extent of atmospheric dispersion. It is not clear from the results presented whether 1) changes in PSNDs in the S-K plane cannot be detected in a conventional parameter space, and 2) any approximate delineations can be proposed that link physical processes to regions in S-K that could demonstrate its usefulness.

---

## Author Comment (AC1) · 26 Oct 2017

**Reply to the comments of Anonymous Referee #3**

We thank Referee#3 for the comments provided to our manuscript. Here we try to reply to the comments at our best, indicating the changes we are going to make in the revised version of the manuscript. With "GC" indicate general comment, while with "SC" specific comment.

GC: "**The authors examine particle size distributions from about 0.3 um to 5 um diameter using two optical particle counters (OPC) located in Milan and on Oga-San Colombano at 2250 msl. They use a skewness-kurtosis plane based on the statistics of the measured size distributions to consider several possible statistical distributions that are commonly used to represent aerosol particle size distributions. They show that the Johnson SB (and SU) are capable of representing relatively complex distributions over the size range of the OPC (0.3-5 um) depending on factors that control total number concentrations, including wind speed and precipitation. This paper offers a new (to me) and interesting approach to contrasting particle size distributions, although it would have been more interesting had the size distributions covered sizes down to 20 or 30 nm, rather than starting at 300 nm. The work appears to be targeted towards models that use modal representations, but the skewness-kurtosis plane with separation by total number concentration seems like it has potential as a tool for analysis of size distributions. The paper is well organized, and the figures are well done, but a careful editing of the paper for grammar is required as misinterpretations are possible. The figure captions could use some additional detail.**"

We thank Referee#3 for your kind and well-focused comment to our manuscript. In this manuscript, we would like to introduce this kind of analyses to PSD data, which could suggest other investigations including other diameter ranges (i.e. 20 or 30 nm) or other pollutants. In the revised version of the manuscript we will make a careful editing of the paper to avoid misinterpretations.

Specific comments:

SC1. "**Page 3, lines 4-5 – You first say the PNSD pattern does not change, and then you say that it varies by site and season. Please clarify. Also, clarify "In this case": winter?**"

Many thanks for this comment. Here we mean that PNSD pattern (i.e. the domain where the sample points are in the skewness-kurtosis plane) is well represented by the Johnson SB domain, except for the urban winter data. So, the PNSD pattern does not change, even if we change the site or the season with the exception mentioned above. We will clarify the point in the revised version of the manuscript in order to avoid misinterpretations.

SC2. "**Page 3, line 10 – What do you mean by "background"? Its meaning in this context needs to be clarified.**"

We use the term "background" to identify a site that is not directly affected by local traffic emissions and is representative of the urban area. To clarify the site characteristics, the paragraph is modified as follows:

"The data have been collected in two sites: the urban site located in Milan at Pascal-Città Studi (45°28'42"N, 9°13'54"E, 120 m a.s.l.) and the rural high altitude site at Oga-San Colombano (46°27'40"N, 10°18'07"E, 2290 m a.s.l.). The urban site is representative of "urban background" conditions and is not direct affected by local traffic emissions (Vecchi et al., 2004)."

The following reference is added:

R. Vecchi, G. Marcazzan, G. Valli, M. Ceriani, C. Antoniazzi, The role of atmospheric dispersion in the seasonal variation of PM1 and PM2.5 concentration and composition in the urban area of Milan (Italy), Atmospheric Environment, 38(27), 2004, 4437-4446.

SC3. "**Page 4, line 1 – What model number? Did you use two counters, one at each site, or was one counter transported between sites? If two counters, were they the same model at both sites? If two counters, how were they compared and validated to determine possible differences associated with the counters rather than the sites? It is not uncommon for these types of counters to have large uncertainty in the smallest nominal size. Was the lowest size evaluated in any way**?"

Many thanks for this comment. The code related to the winter site of Oga San Colombano cited in page 4, line 1, is SC1. We used two code prefixes, "MI" for Milan-Città Studi and "SC" for Oga-San Colombano, followed by a counter (from 1 to 8 for Milan,1 and 2 for San Colombano) which has been associated to the datasets ordered chronologically. This code has been introduced in order to distinguish in an easier way the ten different datasets. The lowest site was 0.3 $\mu g/m3$ for all the datasets.

SC4. "**Page 4, lines 17 and 19 – do you mean ug/m3?**"

Many thanks for this comment. Yes, it is $\mu g/m3$. We fix it in the revised version of the manuscript.

SC5. "**Page 4, line 20 – Similar to 'background', how do you define pristine?**"

We would like to thank the Referee to point out the use of the "pristine" adjective. We apologize for the confusion. As reported in the site presentation, Oga San Colombano is a rural site. We rephrased the sentence as follows:

"The different characteristics of the aerosol number size distributions at Milan (urban site) and Oga-San Colombano (rural site) are shown in Fig. 1."

The adjective pristine is replaced throughout the manuscript.

SC6. "**Page 4, Line 23 – What are low aerosol levels**?"

Thanks for the request. In the revised manuscript we will clarify this issue. We would like to add the mean annual value of PM10 6 $\mu g/m^3$ and its standard deviation 5 $\mu g/m^3$. In addition, we would like to specify that during the summer (JAS) the mean seasonal value of PM10 12 $\mu g/m^3$ and its standard deviation 9 $\mu g/m^3$.

SC7. "**Page 4, lines 25-29 – This tells us nothing other than you have measured some other things (NOx and meteorological quantities). Perhaps that is what you intended, and there is simply a grammar issue? Otherwise, is there a reference for the "influence" investigation? NOx may be considered a component of the aerosol, but it is not an "aerosol compound"**."

Many thanks for this comment. Yes, there is a grammar issue here, we have just measured some other things (NOx and meteorological quantities). We clarify this point in the revised version of the manuscript.

SC8. "**Figure 1 – Why did you choose August 2012 (MI8) rather than August 2011 (MI6) that is at the same time as SC2?**"

We decided to show the results for MI8 because the PNSD samples in MI8 are more in number (41945) respect to MI6 (28715). In this way we considered a greater statistical basis and thus a lower uncertainty in the calculation of the statistical quantities, such as mean and average.

SC9. "**Page 5, lines 6-8 - Figure 1 – The inflections in the distributions between about 0.4 um and 0.8 um are present at both sites, but particularly evident in the Milan results. Rather than "droplet mode particles", the inflections may be a symptom of ambiguities in the scattering function for the particular angle of the OPC, sometimes referred to as Mie ambiguities. Some of the peaks and valleys above 1 um may also be due to this potential problem. The exaggerated inflections in the Milan PNSD relative to the SC PNSD may be due to differences in the index of refraction or the apparently less steep Milan distributions from 0.3 to 0.7 um compared with the SC distributions. You should look at the scattering function versus particle diameter for the counters. If ambiguities are present (i.e. a similar amount of light scattered into the collection angle by particles of different sizes), the common solution is to average across the bins covering the ambiguity size range**."

Many thanks for this comment and for the suggestion. Yes, in the revised version of the manuscript, we removed the ambiguity averaging across the bins. We have produced a new Figure 1.

[Figure]

SC10. "**Figure 2 – Indicate what the lines refer to in the caption**."

Many thanks again for this comment. In the revised version of the manuscript, we will clarify in the caption of Figure 2 that the dotted-dashed line is the lognormal distribution, the dotted line is the Weibull distribution, and the dashed line is the gamma distribution.

SC11. "**Page 7, line 7 and Figure 3 – The fits in Figure 3 are very good considering the detail. Not being familiar with the JSB distribution, I would to see your fitting process discussed in a little more detail, including the Maximum Likelihood Method. The details could be added to the supplement.**

Thanks for this question. The details about the fitting method will be added to the supplement in the revised manuscript.

We would like to add the following text:

"The Johnson SB distribution is a four-parameter distribution characterized by a bounded domain and great flexibility in the shape. These features make JSB applicable to many fields like meteorology (Johnson, 1949; Tang and Lin, 2013), hydrology (Kottegoda, 1987; Wakazuki, 2013) and ecology (Rennolls and Wang, 2005). The four parameters of JSB are calculated here using a Maximum Likelihood method, applied to each minute of the dataset. The parameters are estimated by maximizing the log-likelihood function $L^*$:

$$L^* = N \ln(\gamma) + N \ln(\delta) + N \ln(2\pi)^{-1/2} - N\gamma^2/2 - \sum_{i=1}^{N} \ln(D_i - \xi) +$$

$$-\sum_{i=1}^{N} \ln(\xi + \lambda - D_i) - \gamma\delta\sum_{i=1}^{N} \ln(D_i - \xi) + \gamma\delta\sum_{i=1}^{N} \ln(\xi + \lambda - D_i) +$$

$$-(\delta^2/2)\sum[\ln(D_i - \xi) - \ln(\xi + \lambda - D_i)]^2$$

where $N$ is the sample size. To do this, we used the function "optim" of R language and the iterative process is controlled by setting specified initial values of location and scale parameters and the distribution constraints. In particular, being JSB a bounded distribution in the interval [$\xi$, $\xi+\lambda$] and the DSDs physically bounded by $D_{min}$ and $D_{max}$, the initial values of $\xi$ and $\lambda$ are chosen sufficiently below and above $D_{min}$ and $D_{max}$, respectively. Thus, $\xi_{start}$ is set equal to ($D_{min} - \varepsilon_1$) and $\lambda_{start}$ is set equal to ($D_{max} - \xi_{start} + \varepsilon_2$), where $\varepsilon_1$ and $\varepsilon_2$ are two arbitrarily small quantities. See Cugerone and De Michele, 2015 and Cugerone and De Michele, 2017 for more details. Alternatively, D'Adderio et al. (2016) used a Least Square method applied to theoretical and empirical third order moment to estimate the parameters of JSB."

SC12. **Page 10, lines 10-17 – Higher values of NOx/NO2 may indicate closer temporal proximity to sources. They also suggest the possibility of a greater fraction of particles from primary emissions, but it does not guarantee that primary particle emissions dominate over secondary. Also, it sounds like "These findings support our hypothesis that in urban sites during winter season the increase of primary aerosols emission by local sources causes an evident increase of primary aerosol compounds concentration." is saying that an increase of primary aerosol emissions causes an increase in primary**.

We would like to thank referee #3 to point out the need for clarity at the beginning of page 10. We agree with the referee that the higher NOx to NO2 ratio indicates a closer temporal proximity to sources.

In addition to support our interpretation of such ratio, we report here the results of aerosol chemical composition analysis performed at the Milan urban site during winter 2014 (a field experiment not discussed in the present manuscript). During such experiment we observed that the $NO_2$ to $NO_x$ ratio decreases with the increase of black carbon mass fraction, i.e. a marker of primary emissions. At the same time the $NO_2$ to $NO_x$ ratio increases when the contribution of secondary organic and inorganic aerosol increases (see attached figures).

To clarify the meaning and interpretation of the NOx to NO2 ratio we would like to modify the manuscript as follows:

"It follows that the NO2 to NOx ratio can provide a measure of the oxidative capacity of the atmosphere (Rao and George, 2014; Fernández-Guisuraga et al., 2016) and it's a measure of the temporal proximity to emission sources. In addition, measurement performed in Milan during different field experiments show that NO2 and NOx ratio anti-correlates with black carbon to PM1 ratio, confirming that the NO2 to NOx in urban area is an indicator of the relevance of secondary pollutant formation over primary traffic emissions. In Fig. 4 we have again reported the skewness-kurtosis plane, where we have plotted in black the data points of MI1 (upper panel) and MI2 (lower panel). Then, we have selected the data points belonging to minutes characterized by values of the ratio NOx/NO2 between 1 and 1.1 (red dots – highly oxidizing atmosphere), 1.1 and 1.5 (orange dots – slightly oxidizing atmosphere), 1.5 and 3 (yellow – little oxidizing atmosphere), greater than 3 (green - no oxidizing atmosphere). Both the two datasets are characterized by high aerosol numbers and high percentages of data points outside JSB domain (74 % and 65% respectively). The percentages of data points characterized by a ratio NOx/NO2 greater than 3 are around 50 %, indicating a prevalence of the primary traffic aerosol contribution. If we select only the data points outside the JSB domain, the percentage of data points with ratio greater than 3 (strong prevalence of primary aerosols) is 56 % for MI1 and 50 % for MI2. While, the percentage of data points with ratio greater than 1.5 (light or strong prevalence of primary aerosols) is 88 % for MI1 and 67 % for MI2. These findings support our hypothesis that in urban sites during winter season the increase of primary traffic contributes to the shifts of (β3, β4) couples in the skewness-kurtosis plane."

[Figure]

Figure 1. Dependency of the NO2 to NOx ratio on the chemical composition of submicron aerosol in Milan urban background site during winter 2014. BC/PM1 indicates the black carbon mass fraction, while SIA+SOA/PM1 indicates the secondary inorganic and organic mass fraction (secondary organic aerosol was quantified with positive matrix factorization analysis of organic aerosol mass spectra – data not published).

---

## Author Comment (AC2) · 26 Oct 2017

**Reply to the comments of Anonymous Referee #4**

We thank Referee#4 for the comments provided to our manuscript. Here we try to reply to the comments at our best, indicating the changes we are going to make in the revised version of the manuscript. With "GC" indicate general comment, while with "MC" minor comment.

GC1: "**This manuscript describes the applicability of the Johnson SB distribution for fitting particle size distributions measured by optical particle counters operated at two sites in Italy. The paper first focuses on assessment of the merits of the expression form by examining the fraction of measurements that lie within the region on the skewness-kurtosis plane that is bounded by the Johnson SB envelope of possible solutions. The patterns of data point clusters on that plane are then linked with meteorological and environmental conditions to suggest the more general use in describing the sources and processing responsible for an observed size distribution. The manuscript is reasonably well-written but would require some editing prior to publication**."

We thank Referee#4 for this kind and general comment to our manuscript.

GC2: "**I have identified several specific concerns I have with the manuscript below. More generally though, this simply does not seem to be appropriate for ACP. The dataset described is very limited and rather uninteresting when not complemented by other aerosol and trace gas measurements. More importantly, the dataset is not really the focus of the paper, but rather the technique to describe the dataset is. Thus, in its current form this would be more appropriate for a journal such as AMT. If the authors chose to shift the emphasis more toward the size distributions I still feel that because of the limitations of the dataset this would be better suited for another journal**."

We thank Referee#4 for this comment. Here we want to clarify that in this work we want to propose a new methodology of analysis of PNSD data, based on the skewness-kurtosis plane and the Johnson SB domain, which can be used also to summarize statistically the aerosol dynamics under meteorological conditions. We used two datasets to illustrate the methodology. The methodology is quite general, and with general implications for the assessment of aerosol dynamics. We intend to apply this to other datasets in the near future, as explained in the next point. In this work, we have mainly focused on physical issues, rather than chemical issues, influencing the variability of PNSD. We will investigate chemical issues in a further study. In the revised version of the manuscript, we clarify this issue to improve the presentation of our work. We think that, for the wide breath of the work, ACP is the proper editorial place.

GC3: "**It could be that collaboration with researchers involved in more comprehensive measurement campaigns could be valuable for evaluating the utility of the techniques described here for understanding influences on size distributions. It seems the authors have considerable experience with statistical methods and data analysis, but not with air quality. The relevance of this is that there is far too much text describing rather fundamental details about aerosol sources and sinks and meteorology**."

Thanks for this suggestion. We are planning to continue the collaboration with researchers belonging to Institute of Atmospheric Sciences and Climate –ISAC, National Research Council, Italy. They are regularly involved in comprehensive field campaigns (see e.g. activities documented at the website http://actris-cimone.isac.cnr.it/) measuring aerosol dynamics from the physical and chemical point of views, in addition to meteorological conditions. So, in the near future, we intend to analyze the data of existing comprehensive field campaigns in order to confirm and extend the results obtained in this work.

GC3: "**The averaged and example size distributions shown in Figures 1 and 3 reveal a common characteristic of distributions measured by OPCs - erroneous peaks and troughs that are often linked with features in the scattering intensity vs. size relationship for the optical geometry of the instrument. The fact that they are retained in the distributions suggests the authors didn't invest much time in calibration of the instruments and processing of the data. But more relevant for this paper, those features will influence any fit of the distributions and the location on the S-K diagrams. There is no discussion of these features or their impact**."

We thank Referee#4 for this comment, also pointed out by Referee#3 (see SC9). In the revised version of the manuscript we will fix this issue, i.e., the inflections in the distributions between about 0.4 um and 0.8 um, due to ambiguities in the scattering function for the particular angle of the OPC. We will operate an averaging across the bins, as suggested by Referee#3. So, in the revised version of the manuscript we will report the new figures (1 and 3), and remake the representation of data in the skewness-kurtosis plane.

GC4: "**The authors argue that the Johnson SB distribution is more appropriate for fitting the particle size distributions than more commonly used forms such as the lognormal. But they neglect to discuss the utility of the lognormal because of the direct connection of the parameters describing it with physically meaningful elements of the aerosol distribution (i.e., N, Dp_mean, SD) and the ability to describe variation of those parameters accompanying things such as atmospheric processing. Furthermore, the manuscript largely dismisses lognormals based on the difference between the data points and the single lognormal point on the S-K diagrams. But does the representation as a point presume that only one lognormal is used to fit the distribution? In practice, multiple lognormals are almost always used**."

We thank Referee#4 for pointing at our attention this interesting comment. In the revised version of the manuscript, we would like to address this issue by reporting in the skewness-kurtosis plane, the domain of a mixture of two lognormals (indicated with red dots). According to the OPC size particle classes, a mixture of two distributions is sufficient to keep the modes of the analyzed datasets. We have compared the Johnson SB domain (in dark grey) with the domain of a mixture of two lognormals, as reported here in the figure. This is an original issue never investigated in the literature and we are happy to deal it in the revised manuscript.

[Figure]

In the revised manuscript, we plan to add an appendix where we describe how we have calculated the β3-β4 domain of the mixture. From the figure, it is possible to see that the Johnson SB distribution has a wider domain respect to the mixture of two lognormals, indicating that the Johnson SB distribution is more versatile respect to the mixture of two lognormals in representing the OPC data.

Minor issues:

MC1: "**Page 4, line 1: Grimm model what**?"

The GRIMM model used at Oga San Colombano is "GRIMM 107 Environcheck" as well as at Pascal-Città Studi. We will specify this in the revised manuscript.

MC2: "**Page 4, line 7: What is the basis for the assertion that the composition is different between the two sites. It undoubtedly is, but this still needs some support**."

In the revised version of the manuscript, we will add some support to this sentence. Specifically, we would like to write "Oga San Colombano shows a higher relative contribution of organic aerosol, likely of secondary origin, as suggested by a higher organic to elemental carbon ratio (Sandrini et al., 2014). In addition, Milano shows a higher nitrate to sulfate ratio, in agreement with a stronger impact from combustion sources, such as traffic and industrial emissions (Perrone et al., 2012")."

Perrone M.G. (2012). Sources of high PM2.5 concentrations in Milan, Northern Italy: Molecular marker data and CMB modelling, Science of the Total Environment 414, 343–355.

Sandrini S. et al. (2014). Spatial and seasonal variability of carbonaceous aerosol across Italy, Atmospheric Environment, 99, 587-598.

MC3: "**Page 4, line 26: Nitrogen dioxide and nitric oxide are not aerosol compounds**."

We acknowledge the Referee for pointing out the mistake. The sentence in modified as follows:

"The influence of primary aerosol sources and meteorology on PNSD has been investigated for the site of Milan. To study the effect of pollutant concentration we have used nitrogen dioxide ($NO_2$) and nitrogen oxide (NO) measurements collected with a chemiluminescence technique following the requirements of European Standard EN 14211: 2005: Ambient Air Quality".

MC4: "**Page 6, line 24: Total particle count is meaningless to readers. I assume the authors simply need to divide this by the product of flow rate and sample time to report it in concentration. Additionally, it seems there is confusion about the upper threshold value because it is written both as 10ˆ4 and as 100000 (=10ˆ5)**."

Thank you for the comment. There was an error in Table 2: the threshold value is $10^5$ (100000) and not $10^4$. Regarding the total particle count, we think that this statistical and physical measure is not meaning less, because it allows the readers to have a direct and simple measure of the load of aerosol particles that can be recorded in a minute and to compare the different cases, changing season and/or site.

**MC5:** "**Page 10, top: The NO2 to NOx ratio will be largely dependent on time of day, which will confound the interpretation of its influence on the patterns in the S-K diagrams**."

We agree with the Referee that the NO2 to NOx ratio depends on the time of the day, as primary emissions from traffic do as well. In addition, previous measurements at the urban site here investigated show that the NO2 to NOx ratio anti-correlates with black carbon to PM1 ratio (a marker of primary traffic emissions in this area) and correlates with the ratio of secondary to primary aerosol species (i.e. secondary organic and inorganic aerosol to black carbon plus primary organic aerosol ratio). Unfortunately, during the presented experiment no data on aerosol chemical composition was available to direct evaluate the contribution of primary and secondary components. Thus, we decided to use the NO2 to NOx ratio as a proxy of polluted air mass ageing. To improve clarity, we would like to modify the manuscript as follows:

"It follows that the NO2 to NOx ratio can provide a measure of the oxidative capacity of the atmosphere (Rao and George, 2014; Fernández-Guisuraga et al., 2016). In addition, measurements performed in Milan during different field experiments show that NO2 and NOx ratio anti-correlates with black carbon to PM1 ratio, confirming that the NO2 to NOx in urban area is an indicator of the relevance of secondary pollutant formation over primary traffic emissions.

In Fig. 4 we have again reported the skewness-kurtosis plane, where we have plotted in black the data points of MI1 (upper panel) and MI2 (lower panel). Then, we have selected the data points belonging to minutes characterized by values of the ratio NOx/NO2 between 1 and 1.1 (red dots – highly oxidizing atmosphere), 1.1 and 1.5 (orange dots – slightly oxidizing atmosphere), 1.5 and 3 (yellow – little oxidizing atmosphere), greater than 3 (green - no oxidizing atmosphere). Both the two datasets are characterized by high aerosol numbers and high percentages of data points outside JSB domain (74 % and 65% respectively). The percentages of data points characterized by a ratio NOx/NO2 greater than 3 are around 50 %, indicating a prevalence of the primary traffic aerosol contribution. If we select only the data points outside the JSB domain, the percentage of data points with ratio greater than 3 (strong prevalence of primary aerosols) is 56 % for MI1 and 50 % for MI2. While, the percentage of data points with ratio greater than 1.5 (light or strong prevalence of primary aerosols) is 88 % for MI1 and 67 % for MI2. These findings support our hypothesis that in urban sites during winter season the increase of primary traffic contributes to the shifts of ($\beta_3$, $\beta_4$) couples in the skewness-kurtosis plane."

MC6: "**Figure 3: The use of an unnecessarily large y-axis range obscures the information in the distributions and the quality of the fits**."

Thank you for the comment. We will modify Fig.3 following your suggestion.

[Figure]

MC7: "**Figure 4: The differences among these graphs are pretty modest**."

In our opinion, the difference between the graphs are not so modest. The discrepancies are highlighted by the numbers showed in the legends and by the explanation of Paragraph 4.2.

---

## Author Comment (AC3) · 26 Oct 2017

**Reply to the comments of Anonymous Referee #5**

We thank Referee#5 for the comments provided to our manuscript. Here we try to reply to the comments at our best, indicating the changes we are going to make in the revised version of the manuscript. With "GC" we indicate the general comments.

GC1: "**The manuscript proposes that the skewness-kurtosis plane (the size distribution projected into the third and fourth moments) can be used to follow changes to the particle number size distribution (PNSD), and that four-parameter Johnson SB (JSB) distribution as being sufficient for observing changes to the PNSD in a way that maps to the skewness-kurtosis plane. The authors present PNSDs from four measurement campaigns under different NOx and meteorology conditions. This manuscript includes a few interesting ideas that are less well-known to the broader ACP community, and has potential for novelty and impact**."

We thank Referee#5 for this kind and encouraging comment to our manuscript.

GC2: "**However, at present time the manuscript is strongly recommended for revision and re-submission. The reason for this is recommendation that each of these ideas introduced are not fully developed. As a result, the reader is mostly left with an impression that what is demonstrated is that the size distribution changes when there are changes in meteorology or emission sources, which could be characterized more informatively using traditional approaches (number concentration, modes, etc.)**."

We thank Referee#5 for the comment. In the revised version of the manuscript will improve both the analyses and the presentation. In this manuscript we want to propose a new methodology of analysis of PNSD data, based on the skewness-kurtosis plane and the Johnson SB domain, which can be used also to summarize statistically the aerosol dynamics under meteorological conditions. We want to provide a tool to describe statistically and quantitatively the PNSD variation with meteorological conditions and emission sources. We used two datasets to describe the performance of such a tool. We will improve the presentation of our work, and add additional analyses as explained in the next point.

GC3: "**The conclusion that the JSB can be used to represent PNSDs does not appear to be well-supported by the material that is presented. As one of the other reviewer notes, PNSDs can be multimodal, and representing each of these modes well is in itself a challenge. There is no indication regarding the modality or quality of fit permitted by JSB. What is presented seems to be that the range of skewness and kurtosis in observed PNSDs fall within the range that can be represented by the JSB distribution except at high concentrations. Furthermore, it is not demonstrated that JSB outperforms other parametric distributions for representing PNSDs (except for the reason of having four fitting parameters), and the authors even note in the conclusions that the other parametric representations may be adequate**."

Thanks again for this comment, also raised by Referee#4 (GC4). We know that the lognormal or the mixture of lognormals are generally used in the literature to represent the PNSD data. We agree with the Reviewers that it is a nice idea to compare JSB distribution with those commonly used in the literature, in order to see if the JSB outperforms (or not) other parametric distributions in representing PNSDs. We have analyzed this topic and in the revised version of the manuscript, we would like to address this issue.

Lets consider a mixture of two lognormals. According to the OPC size particle classes, a mixture of two lognormals is sufficient to keep the modes of the analyzed datasets. Let assume that X follows a mixture of two lognormal distributions. Its density $f_X(x)$ is

$$f_X(x) = \sum_{i=1}^{2} \pi_i f_i(x)$$

Where $f_i(x)$ and $\pi_i$ are respectively the (lognormal) density and the weight of the i-th component. The k-th order moment respect the origin of the mixture

$$E[X^k] = \mu^{(k)}$$

has been calculated as

$$\mu^{(k)} = \sum_{i=1}^{2} \pi_i \mu_i^{(k)}$$

where $\mu_i^{(k)}$ is the k-th order moment respect the origin of the i-th component. $\mu_i^{(2)}$, $\mu_i^{(3)}$, $\mu_i^{4)}$ can be written in terms of mean $\mu_i^{(1)}$, standard deviation $\sigma_i$, skewness $\beta_3$ and kurtosis $\beta_4$ of the i-th component as

$$\mu_i^{(2)} = \sigma_i^2 + (\mu_i^{(1)})^2 \quad (1)$$

$$\mu_i^{(3)} = \beta_{3_i}\sigma_i^3 + 3\mu_i^{(1)}\sigma_i^2 + (\mu_i^{(1)})^3 \quad (2)$$

$$\mu_i^{(4)} = \beta_{4_i}\sigma_i^4 + 4\mu_i^{(1)}\beta_{3_i}\sigma_i^3 + 6(\mu_i^{(1)})^2\sigma_i^2 + (\mu_i^{(1)})^4 \quad (3)$$

The skewness $\beta_3$ and kurtosis $\beta_4$ of the mixture are respectively

$$\beta_3 = \frac{\mu^{(3)} - 3\mu^{(1)}\mu^{(2)} + 2(\mu^{(1)})^3}{(\mu^{(2)} - (\mu^{(1)})^2)^{3/2}} \quad (4)$$

$$\beta_4 = \frac{\mu^{(4)} - 4\mu^{(1)}\mu^{(3)} + 6\mu^{(2)}(\mu^{(1)})^2 - 3(\mu^{(1)})^4}{(\mu^{(2)} - (\mu^{(1)})^2)^2}. \quad (5)$$

Substituting Eq.s (1-3) into Eq.s (6,7), it is possible to express skewness and kurtosis of the mixture as a function of mean, standard deviation, skewness and kurtosis of each component.

In the skewness-kurtosis plane, the domain of a mixture of two Lognormal distributions has been determined numerically through a Montecarlo simulation by simulating couples of lognormal distributions with parameter $\mu$ in the range (-10:0.1:10), parameter $\sigma$ in the range (0:0.1:10) and weight $\pi$ in the range (0:0.1:10). The skewness and the kurtosis can be calculated by using equations (4,5).

We compared the Johnson SB domain (in dark grey) with the domain of a mixture of two lognormals (indicated with red dots), as reported here in the figure. This is an original issue never investigated in the literature and we will be happy to deal it in the revised manuscript.

[Figure]

As shown in the figure, the Johnson SB distribution has a wider domain respect to the mixture of two lognormals, indicating that the Johnson SB distribution is more versatile respect to the mixture of two lognormals in representing the OPC data.

GC4: "**Regarding the use of the skewness-kurtosis (S-K) plane, does it provide more information that cannot be achieved by examining other parameters of the PNSD conventionally used (e.g., first and second moments of lognormally transformed data)? Given the long history of modeling PNSDs, the mode gives some indication of whether the dominant source is likely anthropogenic or biogenic; the geometric standard deviation may be related to the extent of atmospheric dispersion. It is not clear from the results presented whether 1) changes in PSNDs in the S-K plane cannot be detected in a conventional parameter space, and 2) any approximate delineations can be proposed that link physical processes to regions in S-K that could demonstrate its usefulness**."

Thank you for this comment, we will improve the description of the skewness-kurtosis plane in the revised manuscript, accordingly.

The S-K has been used here as a powerful and easy-to-use statistical tool for the identification of the statistical distributions which best represent the PNSDs. The peculiarity of the S-K moment-ratio diagram is that every theoretical distribution occupies a specific domain in the plane, that can be a point, a line or an area. Thus, this plane can be used as a diagnostic tool for the identification of distributions able to model given datasets, comparing the domain of the theoretical distributions and the sample variability. For these reasons we can state that the use of this plane provides different information respect the examinations of other statistical parameters, conventionally used for the analysis of PNSDs. In the manuscript we have described our first tentative to study the changes in PNSDs pattern in the S-K plane related to physical processes, in particular the influence of primary and secondary aerosol particles. In the future, we would like to examine more in depth this very interesting and, in our opinion, very promising analysis.

---

## Author Response (AR2)

**Reply to Review Report #1**

The Reviewer says: "(This manuscript shows that the K-S plot can be used to determine the best function form to fit particle number size distribution (PNSD) and to study the evolution of PNSD. This represents a new and interesting approach."

We thank Reviewer #1 for this kind comment.

The Reviewer says: "(1) I share the concern of the other reviewers that the conclusion of this paper is based on quite limited data sets obtained from one instrument."

We agree with you. Clearly this is the first contribution in this direction. We think that others are necessary in the future. Here we want to present the methodology and a first application to PSD data. However, we are confident with our proposal because we have successfully applied it worldwide to Drop-Size-Distribution data, see Cugerone and De Michele 2015, 2017.

The Reviewer says: "(The best function form to describe PNSD likely depends on the sample location, time, and the size range of measured particle size distribution. The size distribution data used in this study also show erroneous peaks and troughs that suggest weaknesses in instrument calibration and/or data processing. I am wondering if the other function forms would fit the PNSD as well as JSB distribution if the erroneous peaks and troughs had been corrected."

In order to remove ambiguities in particle size distributions, as suggested by the Referee, we averaged the particle number across the bins.

In the revised manuscript, we have operated a pre-treatment of data through an averaging across the bins, recalculated the sample skewness and kurtosis and reported these on the S-K plane, without a significant modification of points distribution on the S-K plane.

The advantage of JSB respect to the canonical distributions like Normal, Exponential, Gamma, and Lognormal, is that the latter are represented in the S-K plane through a point (Normal and Exponential) or a line (Gamma and Lognormal), while the former through an area. Looking the sample couples in figures B1, it is possible to see that they occupy almost the entire domain of JSB.

We hope this clarifies the point.

The Reviewer says: " (2) The authors show a point of (0.3) on the SK plot for normal distribution. This suggests that the skewness (zero) is evaluated for the entire size range instead of the size range of the measured PNSD (i.e., 0.3 to 25 microns). For appropriate comparisons, the skewness and kurtosis of fitted functions (e.g., normal, exponential, gamma, lognormal, etc.) should all be calculated using the same size range of the measured PNSD."

Thanks for this comment. To reply, we refer to a previous work by Cugerone and De Michele 2017, where we firstly investigated the S-K plane's domains of upper bounded lognormal and gamma with respect to the domains of the canonical distributions. We showed that the domain of upper bounded lognormal and upper bounded gamma is an area, not a line. See the following figure, and in particular the upper-right panel for the domain of the truncated lognormal and the lower-right panel for the domain of the truncated gamma. We found that the upper bounded lognormal and upper bounded gamma are more suitable to represent the DSD data respect to the canonical lognormal and gamma distributions. Nevertheless, the JSB outperforms the truncated distributions. Since the area occupied by PNSDs in the S-K plane is similar to the area occupied by DSDs we can draw similar conclusions also for PNSDs.

[Figure]

The Reviewer says: " (3) The authors show the PNSD shifts in the S-K plane when primary aerosol emission is strong, and during strong wind or precipitation events. Could the authors provide some explanations on why the PNSD shifts in S-K plane under these conditions?"

Thanks for this comment.

Generally during winter time, we have observed a strong increase of the number of particles with small size, around 0.3-0.4 µm, strongly affected primary sources. This corresponds to a quite strong change in the histogram of particles (basically a peak of frequency in the first diameter classes), which induces a change in the sample skewness and kurtosis, and in the position of the couples ($\beta_3$- $\beta_4$) in the S-K plane. Histograms having such strong peak in the first diameter classes, but with the presence of particles also in the other diameter classes, are generally associated to distributions having a "long" right tail, characteristic of unbounded distributions. Thus, this change tends to move the couple in an area of the S-K plane, which is occupied by the domain of unbounded distributions, such as Johnson SU or generalized hyperbolic distribution.

Conversely, meteorological (rain and wind) events tend to reduce the number of particles in the first diameter classes, having as consequence, that the couples ($\beta_3$- $\beta_4$) tend to come back in the area of the S-K plane, which is occupied by the domain of bounded distributions, JSB.

In the revised manuscript, we clarify this point, see the new figure (Fig. 7) and comments.

**Reply to Review Report #2**

SC3. Were the various counters intercompared at any point, prior to, during or following the ambient measurements? If yes, please indicate in the manuscript.

The counter employed for particles size distribution measurement at San Colombano is the same employed at the urban site. Similar performances of the instrument during the experiments are guaranteed by periodic calibrations.

SC7. Modify your new sentence as "To study the effect of pollutant concentrations here, we use nitrogen …".

We thank the Reviewer#2 for this suggestion. Yes, we did it in the revised manuscript.

SC8. That is fine, but please indicate your reasoning in the manuscript.

The reasons are indicated in section 4.2 of the manuscript. During winter season in Milan we see a significant change of the general PNSD pattern, consisting in a shift toward the center of S-K plane. We found a plausible explanation of PNSD dynamics in the recurrent winter increase of aerosol, due to heating ignition and high traffic levels. In order to clarify this point, we decided to analyze measurements of two common atmospheric components, NO2 and NO which can be connected with this phenomenon. They can indeed be produced by the burning of fossil fuels, like coal, oil and gas.

SC12. Re-write the modified sentence as "In addition, measurements performed in Milan during different field experiments show that the ratio of NO2 to NOx anti-correlates with the ratio of black carbon to PM1, suggesting that the NO2/NOx in these urban areas is an indicator for secondary pollutant formation relative to primary traffic emissions."

We thank the Reviewer#2 for this suggestion. Yes, we did it in the revised manuscript.

I have two suggestions for the Abstract of the revised manuscript:

1) "collected periodically in the urban area of Milan, Italy during 2011…"

2) Remove "mainly belonging to the coarse mode (PMcoarse)". The indicated size range allows the reader to assess what was measured, and the truth of your statement depends on whether you are considering number or mass

We thank the Reviewer#2 for this suggestion. Yes, we have fixed these in the revised manuscript.

**Reply to Review Report #3**

The Reviewer says: "In this revision, the authors have addressed reviewer comments particularly with regards to the explanations of the relationship between aerosol size distributions and NOx/NO2 ratios, and added a relevant and useful illustrations for the mixture of two lognormal modes."
We thank Reviewer #3 for this appreciation.

The Reviewer says: "However, the discussion in Sections 4.2 and 4.3 does not seem to connect with the ACP audience as explanations seem largely correlative. Part of this may be my own unfamiliarity with detecting shifts and interpreting changes in S-K space, but I assume this would reflect upon the wider ACP community for which this topic is also new. What may be helpful would be to expand Figure 4 to connect proposed emission or meteorological factors with changes in individual aerosol size distributions, and then their reduced representation in S-K parameter space; after this the manuscript could be suitable for publication in ACP."
We know that the audience of ACP is not familiar with this type of analysis.

In order to clarify better the evolution of the process, we have produced a new figure (Fig. 7 in the revised manuscript). Here we represent 
[revised manuscript text omitted]